# Peer Support for Public Safety Personnel in Canada: Towards a Typology

**DOI:** 10.3390/ijerph19095013

**Published:** 2022-04-20

**Authors:** Jill A. B. Price, Adeyemi O. Ogunade, Amber J. Fletcher, Rosemary Ricciardelli, Gregory S. Anderson, Heidi Cramm, R. Nicholas Carleton

**Affiliations:** 1Department of Psychology, University of Regina, Regina, SK S4S 0A2, Canada; adeyemi.ogunade@uregina.ca (A.O.O.); nick.carleton@uregina.ca (R.N.C.); 2Department of Sociology and Social Studies, University of Regina, Regina, SK S4S 0A2, Canada; amber.fletcher@uregina.ca; 3Department of Sociology, Memorial University, St. John’s, NL A1C 5S7, Canada; rricciardell@mun.ca; 4Faculty of Science, Thompson Rivers University, Kamloops, BC V2C 0C8, Canada; ganderson@tru.ca; 5School of Rehabilitation Therapy, Queen’s University, Kingston, ON K7L 3N6, Canada; heidi.cramm@queensu.ca

**Keywords:** peer support, public safety personnel, mental health, typology

## Abstract

Public safety personnel (PSP) are frequently exposed to potentially psychologically traumatic events (PPTEs) which can impact mental health. To help mitigate the negative effects of PPTEs, PSP commonly rely on peer support. Peer support generally refers to a wide variety of mental health resources that offer social or emotional assistance to a peer, and in some cases professional assistance. Despite the use of and demand for peer support, there is relatively little empirical evidence regarding effectiveness. The evidence gaps regarding peer support effectiveness may be due, in part, to inadequate guidelines and standards of practice that are publicly supported by a consensus among peer support providers. The current study was designed to explore the current conceptualization and implementation of peer support among Canadian PSP using a document analysis. The results indicate that peer support can be conceptualized via three models (i.e., peer-enabled, peer-led, peer-partnership) and implemented via two delivery methods (i.e., program, service). The research team proposed a novel diagram towards a typology of peer support to highlight the diversity in peer support conceptualization and implementation and provide a foundation for the development of mutually agreed-upon language and a shared framework. Overall, the current study can help inform peer support resources within and beyond PSP communities.

## 1. Introduction

Public safety personnel (PSP) are diverse professionals who work to ensure the safety and security of Canadians. PSP include, but are not limited to, border services officers, correctional workers, firefighters, Indigenous emergency managers, operational intelligence personnel, paramedics, police (federal, municipal, provincial), public safety communicators (e.g., 911 operators, dispatchers), as well as search and rescue personnel [1]. Compared to the general population, PSP are exposed to potentially psychologically traumatic events (PPTEs) at substantially greater frequencies and intensities [2]. A PPTE is a stressful occurrence (e.g., combat, serious transportation accident, sexual assault) that involves direct or vicarious exposure to actual or threatened death, serious injury, or sexual violence and may result in one or more post-traumatic stress injuries [1]. 

Frequent PPTE exposures are inherent to PSP work and, particularly when coupled with other operational stressors, appear to increase risk for developing a myriad of physical and mental health challenges [3,4]. Morbidities that are associated with PPTEs and operational stressors range from diverse physical maladies such as cardiovascular disease, headaches, and kidney stones [4], to mental health challenges such as burnout [5], suicidal thoughts or behaviours [6,7], and mental health disorders (e.g., posttraumatic stress disorder [PTSD]) [8]. Developing and evaluating evidence-based mental health resources to mitigate the effects of PPTEs among PSP should be a priority [8,9]. 

Peer support has been identified as the most common mental health resource that is available to and sought by Canadian PSP to mitigate the effects of PPTE [10]. Despite the demand, there is little empirical evidence to support the effectiveness of peer support on mental health. This lack of empirical evidence has been largely attributed to low fidelity. Fidelity is the extent to which peer support is delivered as originally intended. Providers also vary greatly in terms of peer support duration, timing, content, and outcomes that are measured [9]. The diversity within and across peer support providers has made individual or comparative empirical evaluations extremely challenging [9]. The evaluation of existing approaches is further complicated by a lack of conceptual clarity on what exactly constitutes peer support. These limitations underscore the need to explore the current conceptualization and implementation of peer support in Canada to help identify the best practices. 

Peer support generally refers to a wide variety of mental health resources involving social or emotional assistance to a peer, and in some cases, professional assistance [11]. There are two types of peer support: (a) informal; and (b) formal. Informal peer support might be defined as “less structured support provided by participants who are drawn together by what they have in common, with none more experienced or better prepared to offer support than the other” [12]. Informal peer support interactions can range from social banter to friendly advice via untrained peer supporters on issues that are related to familial, financial, mental, and professional stressors, and are typically derived from friendships that are cultivated over time. Formal peer support might be defined as “support that is offered by trained and/or experienced peer support workers within a structured setting” [12]. Formal peer support typically aims to build resilience, provide psychological first aid, or to train others to provide support. Peer support can then represent a complex social “system of giving and receiving help founded on key principles of respect, shared responsibility, and mutual agreement of what is helpful” [13]. The current study focuses exclusively on formal peer support for Canadian PSP. 

Formal peer support may include a range of resources that are led by trained peer supporters, each with their own training courses, tools, and goals [9]. Peer support training may provide a basic understanding of mental health that is intended to facilitate structured assistance to peers (e.g., intervening after a PPTE, providing psychological first aid, conducting an initial risk assessment to identify a possible mental health injury). The overarching goal of formal peer support among PSP may be to build supportive relationships to reduce stigma, build resilience, and normalize mental health challenges [14]. The goals of peer support may also include, but are not limited to, providing “an empathetic, listening ear; provid[ing] low-level psychological intervention; identify[ing] peers who may be at risk to themselves or others; and facilitat[ing] a conduit for professional help” [15]. Nevertheless, the diversity surrounding formal peer support remains high. The current study was designed to help facilitate cohesion and clarity by exploring how formal peer support is conceptualized and implemented among Canadian PSP. 

## 2. Materials and Methods

The current study presents the findings from one component of a multi-method study of mental health support that is available to Canadian PSP. The first component created a free online repository of mental health resources called PSP Mental Health. The second component was a document analysis of selected PSP peer support providers in Canada. The third component involved an on-going survey of peer support experts (e.g., frontline PSP, PSP leaders, peer support providers, researchers, registered mental health care providers) to validate and obtain their perspectives on our characterization of the conceptualization and implementation of peer support among PSP in Canada and other key issues arising from the second component, such as what constitutes peer support and the ideal elements of effective peer support delivery. This article presents the results of the second component (i.e., document analysis), which employed an inductive, exploratory approach to identify the current conceptualization and application of peer support for PSP in Canada. 

Ethics approval was received from the first author’s University Institutional Research Ethics Board. The research team identified and invited any organization that offered peer support training for Canadian PSP to participate in the study. In total, 32 organizations were identified for the study. The identified organizations were refined based on the following criteria: (a) must offer peer support training (eliminating 1 organization from the sample); (b) must be willing to share their training manual(s) with the research team for the study purposes (eliminating 17 organizations from our sample); (c) manuals must provide sufficient detail surrounding the conceptualization and implementation of their peer support (eliminating 3 organizations from our sample); and (d) peer supporters must have access to one or more registered mental health care providers for referral needs (eliminating 0 organizations from our sample). 

The document analysis examined the textual (i.e., quantitative) and contextual (i.e., qualitative) content within the remaining 11 peer support training manuals (Table 1) [16]. The research team first carried out a text search via a quantitative analysis of the documents for references to keywords such as “peer”, “peer supporter”, and “peer support”, and then coded all references in NVivo 12 analysis software. Next, the coders spread the coding context in NVivo to develop a qualitative understanding of how the keywords were being used in the training manuals. Contextual analysis was used to help identify latent assumptions within the text; for example, who “counts” as a peer and what forms of delivery are promoted. The coders also examined if and how each training manual engaged with evidence of effectiveness; for example, by drawing on empirical literature or conducting fidelity checks. The research team developed a diagram as a first step towards a typology of peer support based on our analysis of the data that were collected. 

## 3. Results

### 3.1. Terminology

Peer-related terms (i.e., peer, peer supporter, peer support) were mentioned in all 11 training manuals a total of 1463 times. The term “peer” was mentioned in all training manuals a total of 903 times (ranging from 1 to 519 terms); however, none of the organizations provided an explicit definition. The term “peer supporter” was mentioned in seven training manuals a total of 240 times (ranging from 1 to 124 terms). Only two organizations provided an explicit definition of peer supporter within their training manuals, which included “[an individual] who understands the impacts of trauma and uses trauma-informed practices will be less likely to unintentionally re-traumatize a peer and more likely to support healing” [17] and “a person who has a mental health concern and might work as a provider of mental health services … [who] has been trained and has the ability to use their recovery experience to help their peers recover” [14]. The diverse definitions of peer supporter highlight the markedly different conceptualizations and the associated potential implications for peer support practices. 

The term “peer support” was mentioned in 10 training manuals a total of 323 times (ranging from 1 to 135 terms). Only five organizations provided an explicit definition of peer support within their training manuals, some of which included “the provision of crisis intervention services by those other than mental health clinicians and directed toward individuals of similar key characteristics as those of the providers” [18] and “the act of people who have had similar experiences with mental health concerns giving each other encouragement, hope, assistance, guidance, and understanding that aids in recovery” [14]. The definitions suggest peer support is currently conceptualized and implemented in different ways among providers in Canada. Peer support can refer to resources that are designed and delivered during acute scenarios to address the immediate impact of PPTE exposures as well as resources that are designed and delivered during non-acute scenarios to address the chronic effects following PPTE exposures. The current results highlight the need for a typology that enhances our understanding of peer support that is available for Canadian PSP.

### 3.2. Typology of Peer Support

The research team developed a diagram towards a typology of peer support to help fill this void (Figure 1). Model refers to a concept of peer support among PSP that can be used for various investigative and demonstrative purposes, such as enhancing understanding of the concept or process, showing relationships, or identifying epidemiological patterns [19]. The document analysis identified three peer support models: (a) peer-enabled; (b) peer-led; and (c) peer-partnership. Peer-enabled represents a model of peer support that is led by external professionals, and assisted by peers, to promote knowledge and skill development that is needed for mental health support among groups of PSP before, during, and/or after a PPTE. Peer-led represents a model of peer support that is led by peer supporters to promote knowledge and skill development that is needed for mental health support among individual or groups of PSP before, during, and/or after a PPTE. Peer-partnership represents a model of peer support that is led by external professionals and peer supporters to promote knowledge and skill development that is needed for mental health support among groups of PSP before, during, and/or after a PPTE. The most common peer support model that was identified by the training manuals in our sample was peer-led (Table 2). 

Delivery refers to diverse approaches to providing peer support to PSP. The document analysis identified two types of peer support delivery: (a) program; and (b) service. A program represents a coordination of two or more peer support services to PSP. A service represents a delivery of peer support to PSP. The evaluation of the training manuals identified ICISF as the only organization offering a peer support program (i.e., CISM), while the other organizations exclusively offer peer support services. A detailed description of the provider delivery options that follows offers evidence supporting our characterization. The most common peer support delivery method that was identified by the training manuals in our sample was service. 

To further explore the nuances of formal peer support, the remaining discourse analysis focused on three organizations that displayed distinct and well-established models of peer support, each of which serves as an exemplar for a particular model of peer support: (a) Wayfound Mental Health Group; (b) the IAFF; and (c) the ICISF. The Wayfound Mental Health Group was selected as it offers a peer-enabled model, the IAFF was selected as it offers a peer-led model, and the ICISF was selected as offers a peer-partnership model. The document analysis focused only on the training manuals as provided by the organizations; therefore, this investigation is limited to the training manual content and does not consider any supplementary material that may exist elsewhere (e.g., organizations’ websites).

### 3.3. Wayfound Mental Health Group

The Wayfound Mental Health Group is an organization that offers a peer support service via resiliency-based tools to mitigate operational stressors among PSP and their families. This service is referred to as BOS and is a 12-month resource consisting of eight modules: (a) establishing the group and operational culture; (b) physiology of stress; (c) markers of stress; (d) cognitive impact; (e) emotions; (f) behavior; (g) communication; and (h) empathy. The modules spend equal time on psychoeducation and group processing. The psychoeducation component is designed to build resiliency by teaching the participants knowledge and skills that are related to stressors. The group processing component is designed to provide the participants with opportunities to understand how to apply tools from psychoeducation and cognitive behavioral therapy within their operational context. The participants also learn how to recognize and respond to operational stressors (e.g., PPTE) in constructive ways by deploying cognitive behavioral therapy skills, as well as functional disconnection and reconnection strategies. Following the eight modules, the participants are provided with 10 monthly follow-up sessions for continued support. 

BOS is delivered entirely by trained, registered, mental health care providers. The training manual emphasizes that BOS is not intended for trauma-processing, despite the therapeutic features within the group processing component. The group processing component promotes six guidelines: (a) confidentiality; (b) equal air-time; (c) non-judgmental approach; (d) timeliness; (e) right to pass; and (f) engagement. The guidelines are designed to provide the participants with controlled opportunities to share experiences that may help others understand and manage the impacts of stressors. In-depth emotional and cognitive processing of stressors is avoided, and participants who require trauma-processing are referred to other clinical resources.

The Wayfound Mental Health Group training manual indicates that peer support is embedded within the group processing component but does not explicitly define peer-related terms. Our analysis suggests that the Wayfound Mental Health Group may define “peer” as another PSP or family member regardless of sector, and “peer supporter” as a trained registered mental health care provider. The term “peer support” may then be represented as a mutual aid between the participants that is delivered by trained peer supporters. Mutual aid is considered a key feature of peer support as a function of providing opportunities for emotional support, information, or guidance that is rooted in shared lived experiences [11]. The group processing component provides the participants with opportunities for emotional information and guidance by means of group discussions and skill development. The manual is currently unclear as to whether the eight modules alone are also considered sufficient to develop the camaraderie that is needed for emotional support amongst the participants. 

The Wayfound Mental Health Group requires peer supporters to complete training and conduct fidelity checks after each module to maintain adherence to the manual. The fidelity checks play an important role in ongoing research as the Wayfound Mental Health Group is one of the few organizations that is currently engaged in empirical evaluations. The manual states that research teams are conducting evaluations with participants prior to beginning and completion of BOS as well as at three-, six-, nine-, and 12-months following completion. None of the results from this empirical research are currently available within the training manual [20]. 

### 3.4. The IAFF

The IAFF is an organization that offers a peer support service that is designed to address and promote behavioral health and wellness among fire service personnel (i.e., firefighters, paramedics, emergency medical technicians) and their families. This service encompasses eight features: (a) active listening; (b) confidentiality; (c) crisis intervention; (d) self-care; (e) assessment; (f) action planning; (g) program planning; and (h) outreach. Active listening consists of verbal and non-verbal communication and is considered by the IAFF to be the cornerstone of peer support to establish trust between the participants. The IAFF suggests that trust and confidence in peer support depends on prioritizing privacy by emphasizing confidentiality. Assessment is required of all the participants to help identify compounding stressors and changes in mood, behavior, and thinking. The IAFF promotes proactive self-care by educating team members on the awareness and recognition of compassion fatigue, vicarious traumatization, and burnout. 

The IAFF designates crisis intervention as an immediate, short-term emotional support while establishing a plan for further ongoing support. Crisis intervention is approached using an ABC model: (a) Approach (i.e., establishing rapport by emphasizing empathetic understanding); (b) Basics (i.e., gathering essential information, identifying, and delineating crises); and (c) Coping (i.e., collaboratively brainstorming potentially helpful coping strategies). The participants are provided with instructions on how to conduct (a) action planning (i.e., collaboration, consultation, referrals, implementation, follow-up, wrap up); (b) program planning (i.e., organization and implementation of peer support); and (c) outreach (e.g., peer-initiated or requested visits, deployment of peers following a PPTE).

The IAFF training manual provides extensive details regarding the recommendations for the development and maintenance of their peer support but does not explicitly define peer-related terms. Our analyses suggest the IAFF may define “peer” as a fire service personnel or their family member regardless of the agency, and “peer supporter” as another trained fire service personnel or their family member. The term “peer support” may then be represented as structured assistance, mutual aid, and social support on a range of issues (e.g., chronic and operational stressors) that is delivered by trained peer supporters who have access to registered mental health care providers. Peer support can also assist with other life events (e.g., divorce, familial issues), but is not intended to replace other mental health resources (e.g., CISM) and should be used in conjunction with registered mental health care providers. 

The IAFF characterization of peer support implies substantial positive engagement from fire service personnel, their leaders, their families, and registered mental health care providers, predicated on delivery by firefighters, paramedics, or emergency medical technicians. Fire service personnel that are interested in facilitating an IAFF peer support are required to complete screening via a free online behavioral awareness course. The online behavioral awareness course is a self-help tool that provides an overview of common behavioral health challenges, operational stressors, and treatment options that are specific to fire service personnel including mental health disorders as defined by the Diagnostic and Statistical Manual of Mental Disorders (DSM-5). Anyone who passes the behavioral awareness course screening is invited to complete the training course. The IAFF training manual does not mention any use of fidelity checks among their peer supporters nor any empirical research supporting the effectiveness of their peer support materials [21]. 

### 3.5. The ICISF

The ICISF is an organization that offers a peer support program via a system of specialized crisis interventions that are available to PSP and their families. This program is collectively referred to as CISM and consists of six core components: (a) assessment and triage; (b) strategic planning; (c) individual crisis intervention; (d) informational group processes; (e) interactive group processes; and (f) resiliency. These core components are designed to provide a comprehensive, integrated, systematic, and multi-component system of crisis interventions that can promote individual and group resiliency before, during, or after a PPTE exposure. Assessment and triage are used to help identify the most appropriate tools by careful evaluation of the PPTE and the individuals that are impacted. Strategic planning provides information that is necessary for developing and implementing operational strategies (e.g., selection of peer supporters and timing of tools), and for avoiding using inappropriate techniques. 

CISM offers peer support in individual and group settings. The ICISF considers individual crisis intervention (e.g., SAFER-R) to be a form of psychological first aid that should be used as a one-time CISM tool in collaboration with other mental health resources. The individual crisis interventions are designed to deliver information and instructions to one individual after a PPTE exposure. Group crisis interventions consist of informational and interactive group processes that are available after a PPTE exposure. The informational processes (e.g., rest, information, and transition services, crisis management briefing) are designed to deliver information and instructions to heterogeneous groups (i.e., individuals without a current relationship, shared history, or common lived experiences). The interactive processes (e.g., critical incident stress debriefing, defusing) are designed to deliver information and facilitate discussion among homogeneous groups (e.g., similar experience, age, gender, rank, family structure). The ICISF training emphasizes important differences between the informational and interactive group processes, and cautions against using interactive group processes with heterogeneous groups. 

The individual and group crisis interventions are designed to build personal and community resiliency. The ICISF defines resiliency by delineating resistance, resilience, and recovery. Resistance is considered the foundation of resiliency and refers to the ability to resist distress, impairment, and dysfunction, before and during a PPTE exposure. When resistance fails, CISM turns to resilience for effective coping strategies. Resilience is the ability to rapidly rebound after or recoil from distress and rise above adversity during and after a PPTE exposure. If resilience deteriorates, mental health challenges may arise that now require recovery efforts. Recovery is the ability to resolve, repair, reconstruct, restore, and rebuild mental health after a PPTE exposure. 

The ICISF training manuals provide extensive detail regarding the services that are offered under their CISM program, but do not explicitly define all peer-related terms. Our analysis suggests that the ICISF may define “peer” as a PSP or family member regardless of their agency, and “peer supporter” as a trained registered mental health care provider, chaplain, PSP, or family member. The ICISF defines “peer support” as a collection of crisis intervention services between participants that are delivered by trained peer supporters. The manual elaborates that peer support is a collection of crisis intervention services that are “most effective” when run and staffed by an established and functioning peer support resource that is internal to the PSP agency where the PPTE exposure occurred. In the absence of an internal peer support team, CISM can be provided via an external entity; however, development of an internal peer support team should be prioritized after the acute phase of the PPTE exposure. Therefore, CISM can be delivered either internally or externally to the specific PSP agency, with a preference for internal delivery, and teams must include a registered mental health care provider who collaborates with the other peer supporters. 

The roles and responsibilities of CISM team members include helping others to: (a) lower emotional tension, stabilize the individual, mobilize an individual’s resources, and mitigate the impact of the PPTE; (b) normalize an individual’s reactions and facilitate normal recovery processes; (c) restore individuals to adaptive functions and enhance unit cohesion as well as unit performance in homogeneous groups; and (d) identify and refer individuals who may need additional care to registered mental health care providers. All team members are also required to complete CISM training in support of fidelity to the ICISF standards of practice. Missing from the ICISF training manuals are references to current peer-reviewed research providing empirical evidence of the overall effectiveness, or of the effectiveness of CISM-trained team members relative to individuals with other training who provide information or interventions after a PPTE exposure [19,22].

## 4. Discussion

The current study explored the conceptualization and implementation of peer support among Canadian PSP. A document analysis of 11 peer support training manuals suggests that the term “peer” ultimately appears to be idiosyncratically defined; however, a peer might be someone with any or all of the following characteristics: (a) shared lived experience; (b) same or a similar broad professional category (e.g., PSP); (c) same or a similar PSP sector (e.g., police); (d) same PSP agency (e.g., Regina Police Service); or (e) shared demographics (i.e., “homogeneity”) that are deemed important to the individual receiving support (e.g., similar experience, age, gender, rank, family structure). The requirements for someone to be perceived as a peer may also change depending on the contextual variables that are relevant to any specific PPTE and may be flexible depending on who is available to provide support when needed. The term “peer supporter” might then include (a) trained PSP or family member; (b) registered mental health care provider; or (c) chaplain.

The term “peer support” can be broadly defined as a supportive relationship between peers and peer supporters; however, each organization appears to conceptualize the operational specifics of peer support in distinct ways. A singular and broad classification of peer support may then mask important multidimensional opportunities to improve PSP mental health. The research team, therefore, proposes a novel and inclusive diagram towards a typology of peer support (Figure 1). Our analysis identified three unique models of peer support: (a) peer-enabled; (b) peer-led; and (c) peer-partnership. Each model varies depending on the role of peers in providing support and can then be implemented via two delivery methods: (a) program; or (b) service. 

The research team further explored the nuances of formal peer support by investigating three organizations that displayed distinct and well-established models of peer support: (a) the Wayfound Mental Health Group; (b) the IAFF; and (c) the ICISF. The Wayfound Mental Health Group offers a peer support service (i.e., BOS) that delivers a peer-enabled model via registered mental health care providers. Peer support is embedded within the group processing sessions through mutual aid between the participants. Eligible participants include PSP and their families; however, the training manual did not explicitly state whether families would be supported independently of the PSP, or only be supported as a function of PSP receiving peer support. The Wayfound Mental Health Group requires that their peer supporters complete training and conducts fidelity checks after each module to ensure adherence to their manual. The fidelity checks are highlighted as providing important data for ongoing empirical evaluation [23]. Following the eight modules, the participants are provided with 10 monthly follow-up sessions for continued support with a registered mental health care provider. The participants who require additional support can be referred to other registered mental health care providers for next-level-of-care.

The IAFF offers a peer support service that delivers a peer-led model via fire service personnel. Eligible participants include fire service personnel (i.e., firefighters, paramedics, and emergency medical technicians) and their families; however, the IAFF training manual did not explicitly state whether families would be supported independent of the fire service personnel, or only be supported as a function of fire service personnel receiving peer support. The IAFF requires that their peer supporters complete training to support adherence to their manual; however, the manual does not mention use of fidelity checks among their peer supporters and does not provide references to peer-reviewed empirical research supporting the effectiveness of their model. The participants who require additional support can be referred to registered mental health care providers for next-level-of-care. 

The ICISF offers a peer support program (i.e., CISM) that delivers a peer-partnership model via a team of registered mental health care providers, chaplains, and PSP. Eligible participants include PSP and their families. The training manuals also highlight resources (e.g., family crisis intervention) that are available to family members independent of their PSP. The ICISF requires that their peer supporters complete CISM training to support adherence to their manual; however, the manual does not mention use of fidelity checks among their peer supporters and does not provide references to peer-reviewed empirical research supporting the effectiveness of their model. Participants who require additional support can be referred to registered mental health care providers for next-level-of-care.

The proposed diagram towards a typology of peer support was designed to highlight the diversity among peer support resources that are available to Canadian PSP using consistent and structured language, as part of supporting steps towards shared language for a community of practice. Shared language can benefit organizations by clarifying conceptualizations and implementations of peer support and can benefit individuals by clarifying their options for peer support and supporting choices that are based on individual preferences. The diagram can further be adapted to include new conceptualizations and implementations of peer support in the future. 

## 5. Future Research

The current relative dearth of peer support research may be inhibiting access to an important and sought-after avenue for supporting mental health among PSP and their families [10]. There is a substantial need for researchers to examine the effectiveness of peer support models, programs, services, and content that are offered in Canada. Train-the-trainer courses offer promising avenues for initial research and may benefit from increasing use of shared language such as that which is provided by the Canadian Institute for Public Safety Research and Treatment Glossary [1]. Cross-sectional and longitudinal research can be conducted to assess, compare, and contrast the impact of different peer support models, programs, services, and content. 

Fidelity of peer support delivery can also be investigated. The lack of current organizations offering fidelity checks, as highlighted by this investigation, is concerning as accountability and adherence to the training as designed is ambiguous. Independently developed research results can further help to develop tailored evidence-based best practices and iteratively improve peer support. In the absence of empirical evidence, there is no way to understand what variables are changing as a result of the peer support, the size of the changes, or how long the changes last. Organizations are encouraged to explicitly include citations and references for any empirical evidence supporting their training in their manuals, and to follow-up with peer supporters who successfully complete their training to ensure that peer support is being delivered as intended. 

Future research would further benefit from the development and maintenance of a national peer support community of practice that is represented by a multi-disciplinary committee that includes PSP, peer support providers, peer support researchers, and registered mental health care providers. The national community of practice and the associated committee could support the (a) standardization of the peer support elements using tools such as the proposed diagram towards a typology; (b) iterative development, revision, and promotion of peer support guidelines; (c) the development of national standards of peer support practice; and d) the development of an accepted external entity that is responsible for the regulation and administration of peer support designations (i.e., accreditation and certification). 

## 6. Conclusions

The current study explored the conceptualization and implementation of peer support that is available to Canadian PSP. The document analysis focused on organizations who self-identified as offering peer support to Canadian PSP. The results suggest that “peer” and “peer supporter” are ultimately idiosyncratically and contextually defined, and that “peer support” is a supportive relationship between peers and trained peer supporters; however, the models, programs, services, and content that are associated with peer support appear distinct across organizations, with potentially impactful differences. The research team proposed a novel diagram towards a typology of peer support (Figure 1) to provide clarity regarding the conceptualization and implementation of formal peer support. The typology diagram may further offer providers, users, and researchers with mutually agreed upon language and a shared framework.

## Figures and Tables

**Figure 1 ijerph-19-05013-f001:**
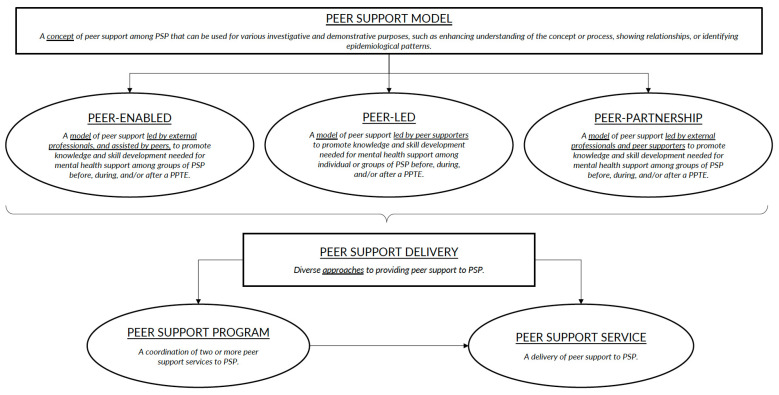
A diagram towards a typology of peer support.

**Table 1 ijerph-19-05013-t001:** List of participating organizations and their respective peer support training.

Organization	Peer Support Training Manual
911 Resilience	Peer Support Training
Canadian Mental Health Association	Operational Stress Injury Canada (OSI-CAN)
Canadian Mental Health Association	Resilient Minds
Department of National Defence	Road to Mental Readiness
The International Association of Fire Fighters (IAFF)	Peer Support Training
The International Critical Incident Stress Foundation (ICISF)	Critical Incident Stress Management (CISM)
Mood Disorders Society of Canada	Peer and Trauma Support Systems
Prairies to Peaks Consulting Inc.	Peer Specialist Training
Tend Academy Ltd.	Train the Trainer Program
Wayfound Mental Health Group	Before Operational Stress (BOS)
Wounded Warriors Canada	Trauma and Resiliency Program

**Table 2 ijerph-19-05013-t002:** Models of peer support as delivered by each organization.

Peer-Enabled	Peer-Led	Peer-Partnership
Tend Academy Ltd.	911 Resilience	The ICISF
Wayfound Mental Health Group	Canadian Mental Health Association (OSI-CAN; Resilient Minds)	
Wounded Warriors Canada	Department of National Defence	
	The IAFF	
	Mood Disorders Society of Canada	
	Prairies to Peaks Consulting Inc.	

## Data Availability

The data that are presented in this study are available on request from the corresponding author.

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
