# Peer review of "Peer Support for Public Safety Personnel in Canada: Towards a Typology"

_ijerph, 2022, doi:10.3390/ijerph19095013_

Round 1

Reviewer 1 Report

Dear authors,

If I understood correctly, you have established a typology for peer support with three types using only the information about three organizations. That would be a bit forced. In order for the three types to be considered a typology, you should show that the support provided in the other identified organizations is also one of three, or a combination of these.
Otherwise, it is just a presentation of three types of peer support, not a typology.
If you want to keep the idea of typology, then show how you identified the 32 organizations, why you consider them representative and where each fits in the typology.
If you only want to present the three distinct types, then insist on the advantages and disadvantages of each, thus comparing their efficiency. You can present them as part of a future typology, which will capitalize on future research on the diversity of peer support variants.

I consider it necessary to redo / develop the article in one of these directions, or in another, which will give value to the research results.

Author Response

Thank you for the feedback. Upon reflection, we agree that more information was required in our methodology and results. The research team has thus provided clarification on how each training manual acquired for our analysis falls into the proposed models of peer support. We have also changed the language throughout the manuscript from "typology of peer support" to "diagram towards a typology of peer support" to reinforce that this is simply a proposal based on our sample; which is flexible and subject to change with subsequent research. 

Reviewer 2 Report

The topic is timely and this paper may ultimately represent a useful contribution to the literature.  However, there are some concerns at this point. 

Although qualitative analysis is mentioned, there is not much 'analysis' present - the work at this point is more descriptive.  The problem stems from a lack of detail about the analysis conducted.  You may want to consult Creswell, or Miles, Huberman, & Saldana, about how to write up qualitative analysis based on the coding you have employed.  There needs to be more clarity so that the raw data can be shown to closely match the resultant themes/codes.  It does seem like an analysis was conducted (given the discussion) - but this could be made clear through presentation of information about the codes that resulted, and their distribution among the works in the corpus. 

Second, there is a lot of attention to the individual works, which is perhaps understandable given that there are so few of them, but more effort to draw conclusions from the corpus as a whole would be helpful.

Would the corpus be made available from the authors on request, given that the information is not (apparently) readily obtained?
Are there examples from the texts that can be provided to illustrate points more fully?
The topic is interesting and more research is welcome in the area, but the manuscript is, at this point, underdeveloped.

Author Response

Thank you for the insightful feedback.

  1. You are correct. This manuscript utilized both qualitative and quantitative analyses. The research team has updated the manuscript to reflect how each of these analyses contributed to our methodology and results. We have also supplemented the results with data of these qualitative and quantitative analyses based on our sample.
  2. We agree that more attention was needed from the corpus as a whole. As such, we supplemented the results with data surrounding the use of peer support terminology as well as the conceptualization and implementation of peer support. This data, which was pulled from the sample as a whole, helps to support our findings for lack of common language and framework within the peer support community. 
  3. No, training manuals received for this project were obtained via non-disclosure agreements (NDAs).

Round 2

Reviewer 1 Report

Dear authors,
I appreciate the changes made. I consider that the article can be published.